# Synthesis, Characterization, Hirshfeld Surface Analysis, Crystal Structure and Molecular Modeling Studies of 1-(4-(Methoxy(phenyl)methyl)-2-methylphenoxy)butan-2-one Derivative as a Novel α-Glucosidase Inhibitor

Chandra Shivanna [1,†], Shashank M. Patil [2,†], C. Mallikarjunaswamy [3], Ramith Ramu [2,*], Prabhuswamy Akhileshwari [4], Latha Rani Nagaraju [4], Mandayam A. Sridhar [4], Shaukath Ara Khanum [5], V. Lakshmi Ranganatha [6,*], Ekaterina Silina [7], Victor Stupin [8] and Raghu Ram Achar [9]

[1] Department of Physics, The National Institute of Engineering, Manandavadi Road, Mysore 570008, India; chandru@nie.ac.in
[2] Department of Biotechnology and Bioinformatics, JSS Academy of Higher Education and Research, Mysuru 570015, India; shashankmpatil@jssuni.edu.in
[3] Postgraduate Department of Chemistry, JSS College of Arts, Commerce and Science, JSS Research Centre (A Recognized Research Centre of University of Mysore), Mysuru 570025, India; mallik.aanekere@gmail.com
[4] Department of Studies in Physics, University of Mysore, Manasagangotri, Mysuru 570006, India; akhila@uomphysics.net (P.A.); rani04latha@gmail.com (L.R.N.); mas@physics.uni-mysore.ac.in (M.A.S.)
[5] Department of Chemistry, Yuvaraja's College, University of Mysore, Mysore 570005, India; shaukathara@yahoo.co.in
[6] Department of Chemistry, The National Institute of Engineering, Manandavadi Road, Mysore 570008, India
[7] Department of Human Pathology, I.M. Sechenov First Moscow State Medical University (Sechenov University), 119991 Moscow, Russia; silinaekaterina@mail.ru
[8] Department of Hospital Surgery 1, N.I. Pirogov Russian National Research Medical University (RNRMU), 117997 Moscow, Russia; stvictor@bk.ru
[9] Division of Biochemistry, School of Life Sciences, JSS Academy of Higher Education and Research, Mysuru 570015, India; rracharya@issuni.edu.in
* Correspondence: ramith.gowda@gmail.com (R.R.); lranganath.v@nie.ac.in (L.R.); Tel.: +91-09-98-6380-920 (R.R.); +91-09-88-0680-493 (L.R.)
† These authors contributed equally to this work.

**Abstract:** The crystal compound was synthesized and characterized using conventional analytical techniques. The compound $C_{19}H_{21}O_3$ crystallizes in a monoclinic crystal system with the space group *P21/c*. The crystal structure is stabilized by C-H . . . O interactions. The structure is further reinforced by π-π interactions. During in vitro inhibition of α-glucosidase, the crystal compound exhibited a significant inhibition of the enzyme ($IC_{50}$: 10.30 ± 0.25 μg/mL) in comparison with the control, acarbose ($IC_{50}$: 12.00 ± 0.10 μg/mL). Molecular docking studies were carried out for the crystal compound with the α-glucosidase protein model, which demonstrated that the crystal molecule has a good binding affinity (−10.8 kcal/mol) compared with that of acarbose (−8.2 kcal/mol). The molecular dynamics simulations and binding free energy calculations depicted the stability of the crystal molecule throughout the simulation period (100 ns). Further, a Hirshfeld analysis was carried out in order to understand the packing pattern and intermolecular interactions. The energy difference between the frontier molecular orbitals (FMO) was 4.95 eV.

**Keywords:** crystal structure; Hirshfeld surfaces; α-glucosidase inhibition; molecular docking simulation; molecular dynamics simulations; binding free energy calculations

## 1. Introduction

Diabetes mellitus is a common chronic metabolic condition that causes high blood sugar levels due to damage to the specialized cells (islets of Langerhans) that produce insulin in the human body. Diabetic individuals either do not produce enough insulin

or cannot use it effectively [1,2]. Hyperglycemia is caused by a malfunction that causes high blood glucose levels in the body. Diabetes mellitus has two pathways, with type 2 (non-insulin dependent) diabetes mellitus being more common than type 1 (insulin dependent) [3]. The enzyme α-amylase is linked to diabetes type 2 in a direct way. α-Amylase is a pancreatic and salivary gland secretory substance that hydrolyzes complex carbohydrates into polysaccharides, most commonly starch to glucose and maltose in the intestine. By the action of α-glucosidase, they are further degraded to monosaccharides and released into the bloodstream, raising blood sugar levels [4,5].

Instant hydrolysis of carbohydrates can be slowed by limiting the actions of α-amylase, which controls the quick rise in blood sugar levels [6]. The current medication options for amylase and glucosidase have a number of adverse effects that limit their utility in diabetic treatment. As a result, alternative medicines with low side-effects are urgently needed to act as an option to the treatment of diabetes mellitus [7].

Phenoxyacetates are very robust moieties in the face of various harsh reaction conditions. Phenyl acetate is an aromatic fatty acid metabolite of phenylalanine with potential antineoplastic activity, and its stability is documented by numerous transformations on the aryl system without affecting the side chain [8]. Phenoxyacetic acids are very important chemicals due to their wide distribution and extensive use as plant growth regulators, and they are employed on a large scale for weed control on cereal crops and lawns. Phenoxyacetic acid induces hematopoietic cell proliferation, providing potential for oral therapeutics. Particularly, ethyl phenoxyacetate and its derivatives exhibit potential anti-inflammatory and plant growth regulation activity [9]. Modification of the oxyacetamideureido-phenyl moieties of compounds in the phenoxy acetic acid series is considered likely to lead to more potent antagonists. Therefore, phenoxyacetic acid analogues are interesting to study by various chemical and physical means, as these derivatives are very useful in hyperglycemia and insulin resistance treatment [10–12].

In view of their broad spectrum of important medicinal applications and as a part of our ongoing research on the synthesis and characterization of novel compounds, the title molecule 1-(4-(methoxy(phenyl)methyl)-2-methylphenoxy)butan-2-one was synthesized. Herein we report on the crystal structure, Hirshfeld surface analysis, and molecular modeling studies.

## 2. Materials and Methods

### 2.1. Chemicals and Instrumentation

All chemicals were purchased from Sigma-Aldrich Chemicals Pvt Ltd., St. Louis, Missouri, United States, and all are analytical grade 99.0% pure. Melting points were determined on an electrically heated VMP-III melting point apparatus. The FTIR spectra were documented using KBr discs on a FTIR Jasco 4100 infrared spectrometer. The 1H NMR spectra were recorded using a Bruker DRX 400 spectrometer at 400 MHz with TMS as an internal standard. Mass spectra were recorded on a LC-MS/MS (API-4000) mass spectrometer. An additional elemental analysis of the compound was performed on a Perkin Elmer 2400 elemental analyzer.

### 2.2. Synthesis of 1-(4-(Methoxy(phenyl)methyl)-2-methylphenoxy)butan-2-one (3)

1-(4-(Methoxy(phenyl)methyl)-2-methylphenoxy)butan-2-one (3) was attained by refluxing a mixture of 4-(methoxy(phenyl)methyl)-2-methylphenol (0.01 mol) and 1-chlorobutan-2-one (2) (0.02 mol) in dry distilled acetone (75 mL) and anhydrous potassium carbonate (0.02 mol) for 4 h. The reaction mixture was cooled to room temperature and the solvent was removed using a flash evaporator. The residual mass was triturated with ice-cold water to remove potassium carbonate and extracted with ether (3 × 50 mL). The ether layer was washed with a 10% sodium hydroxide solution (3 × 50 mL), followed by water (3 × 30 mL), and then dried over anhydrous sodium sulphate and evaporated to dryness to obtain a crude solid. Further recrystallization with ethanol afforded the title compound in a pure state (Figure 1).

**Figure 1.** Synthesis of the crystal molecule.

1-(4-(Methoxy(phenyl)methyl)-2-methylphenoxy)butan-2-one (3): Yield 75%. M.p.80–82 °C; IR (KBr): 1745 (C=O), 1510–1620 (aromatic) 2860–2800 cm$^{-1}$ (O-CH$_3$). 1H NMR (CDCl3): δ 1.3 (t, J = 6 Hz, 3H, CH$_3$ of ester), 2.3 (s, 3H, Ar-CH$_3$), 2.55 (q, J = 6 Hz, 2H, CH$_2$), 3.25 (s, 3H, OCH$_3$), 4.35 (s, 2H, OCH$_2$), 5.45 (s, 1H, Ar-CH-O), 7.2–7.8 (bm, 8H, Ar-H). Mol. Wt.: 298.38 (M + 1): 299.16 (100.0%), Anal. Cal. for C19H22O3 (298.38): C, 76.48; H, 7.43. Found: C, 76.45; H, 7.47%.

### 2.3. X-ray Crystallographic Analysis

A single crystal with of appropriate dimensions was selected for X-ray diffraction analysis. Data were collected using a Bruker Kappa Apex II Single Crystal X-ray Diffractometer equipped with Cu Kα radiation and a CCD detector [13]. The crystal structure was solved and refined by using SHELXS/L-18 software [14]. The obtained model was refined by isotropic thermal parameters, and later by anisotropic thermal parameters. The geometric calculations were carried out using the program PLATON [15]. The molecular and packing diagrams were generated using Mercury CSD 2.0 [16].

### 2.4. α-Glucosidase Inhibition Assay and Kinetics

The inhibition test for yeast α-glucosidase (EC 3.2.1.20, a type-1 α-glucosidase, was conducted as defined earlier [3]. The inhibitory activity of the test compound was represented by the least-squares regression line of logarithmic concentrations plotted against percentage inhibition, which yielded the IC$_{50}$ values (μg/mL). When compared to the control, this number (IC$_{50}$ values) shows the concentration of samples that can inhibit

enzyme activity by 50%. The inhibition kinetics of the compound against $\alpha$-glucosidase was determined using the method described by Maradesha et al. [17].

### 2.5. Molecular Docking Simulation

The protein sequence of *Saccharomyces cerevisiae* $\alpha$-glucosidase MAL-32 obtained from UniProt (UniProt ID: P38158) was used to construct a homology model using SWISS-MODEL. The model was constructed using the X-ray crystal structure of *S. cerevisiae* isomaltase (PDB ID: 3AXH), which revealed a 72% identical and 84% similar sequence at a resolution of 1.8 Å. Construction of this protein model was essential, as the human $\alpha$-glucosidase protein is yet to be characterized. Since the authors used the yeast $\alpha$-glucosidase in the in vitro studies, homology model of *S. cerevisiae* $\alpha$-glucosidase MAL-32 from UniProt was constructed. Protein and ligand preparation was performed according to the previous study conducted by Patil et al. [18]. Since the constructed model had already been evaluated in previous work by the authors [17], the same model was used in the present study. The binding site prediction and positioning of the binding pocket was established according to the previous work of the authors [18]. The binding residues were placed in a grid box measuring 30 Å $\times$ 30 Å $\times$ 30 Å positioned at the coordinates x = −17.489 Å, y = −8.621 Å and z = −19.658 Å using the software AutoDock Tools 1.5.6. For molecular docking simulation, the protein and ligand preparations were undertaken according to the previous work by the authors [18] using AutoDock Tools 1.5.6 software. The ligand molecule was docked into the protein target using the software AutoDock Vina 1.1.2. Acarbose was used as a control [19].

### 2.6. Molecular Dynamics Simulation

A command-line interface software package, GROMACS-2018.1, was used to perform molecular dynamics (MD) simulation. It is specifically designed for biochemical molecules such as proteins, lipids, and nucleic acids that possess a great many complex bonded interactions. For systems with hundreds to millions of particles, the program can simulate the Newtonian equations of motion, as well as calculate nonbonded interactions swiftly. Based on the previous study conducted by Patil et al. [20], docked complexes of $\alpha$-glucosidase protein with the crystal compound, as well as acarbose with the most negative binding affinities, were submitted for simulation. The simulation boxes, consisting of a protein–crystal compound (9463 residues) complex and a protein–acarbose (9472 residues) complex, were simulated for 100 ns at a temperature of 310 K and 1 bar pressure [19]. A trajectory analysis of root-mean-square deviation (RMSD), root-mean-square fluctuation (RMSF), radius of gyration (Rg), ligand–hydrogen bonds, and solvent-accessible surface area (SASA) parameters was performed and the results plotted using XMGRACE, a GUI based software for plotting the results of MD simulation [19,20].

### 2.7. Binding Free Energy Calculations

Using the MD simulation results, both protein–ligand complexes were subjected to binding free energy calculations using the Molecular Mechanics/Poisson–Boltzmann Surface Area (MM–PBSA) technique. This is an efficient and reliable free energy simulation method used to model molecular recognition, such as for protein–ligand binding interactions. A GROMACS program, g_mmpbsa [21] with the MmPbSaStat.py [22] script was exploited to evaluate the binding free energy for each protein–ligand complex. The g_mmpbsa program calculates binding free energy using three components: molecular mechanical energy, polar and apolar solvation energies, and molecular mechanical energy. The binding free energy was computed using the molecular dynamics trajectories of the last 50 ns and dt 1000 frames. Equations (1) and (2) were used to calculate the free binding energy [23,24].

$$\Delta G_{Binding} = G_{Complex} - (G_{Protein} + G_{Ligand}) \tag{1}$$

$$\Delta G = \Delta E_{MM} + \Delta G_{Solvation} - T\Delta S = \Delta E_{(Bonded\ +\ non\text{-}bonded)} + \Delta G_{(Polar\ +\ non\text{-}polar)} - T\Delta S \tag{2}$$

$G_{Binding}$: binding free energy; $G_{Complex}$: total free energy of the protein–ligand complex; $G_{Protein}$ and $G_{Ligand}$: total free energies of the isolated protein and ligand in solvent, respectively; $\Delta G$: standard free energy; $\Delta E_{MM}$: average molecular mechanics potential energy in vacuum; $G_{Solvation}$: solvation energy; $\Delta E$: total energy of bonded as well as non-bonded interactions; $\Delta S$: change in entropy of the system upon ligand binding; T: temperature in Kelvin [25,26].

## 3. Results and Discussion

### 3.1. X-ray Crystallographic Details

The asymmetric crystal structure consists of two molecules (A and B). Visualization (ORTEP and packing) of the synthesized molecule has been given in Figure 2. Crystallization data and structure refinement of the crystal molecule have been detailed in Table 1.

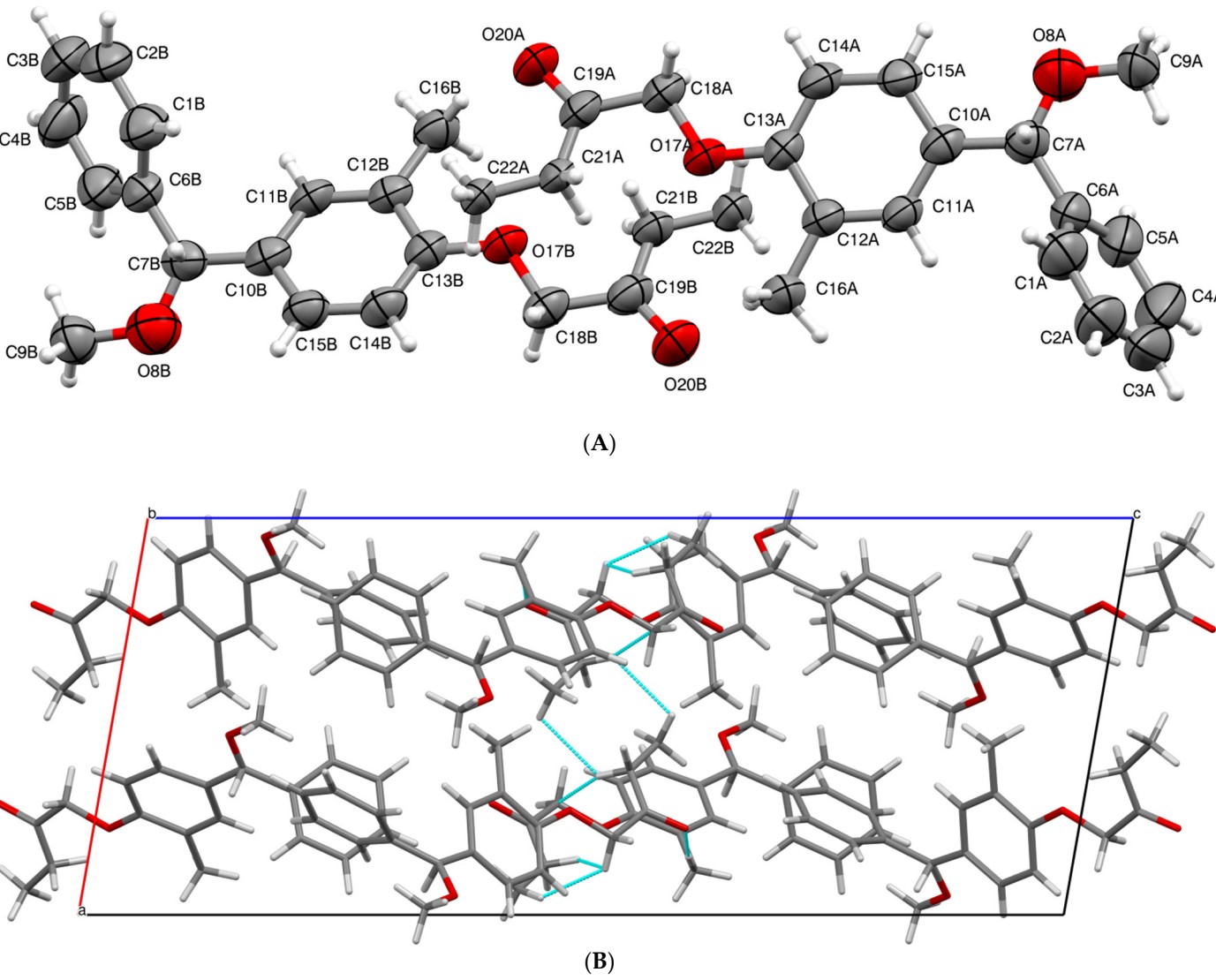

(**A**)

(**B**)

**Figure 2.** (**A**) ORTEP of the molecule (3) with thermal ellipsoids drawn at 50% probability; (**B**) Packing diagram of the molecule (3).

**Table 1.** Crystal data and structure refinement for 3.

| Parameter | Data |
| --- | --- |
| Identification code | 3 |
| Empirical formula | $C_{19}H_{22}O_3$ |
| Formula weight | 298.36 |
| Temperature | 296 K |
| Wavelength | 1.54178 Å |
| Crystal system, space group | Monoclinic, *P21/c* |
| Unit cell dimensions | a = 11.1757(2) Å, b = 10.2148 Å |
| | c = 27.0700(9) Å, β = 99.853° |
| Volume | 3044.66(17) Å$^3$ |
| Z, Calculated density | 8, 1.302 Mg/m$^3$ |
| $F_{000}$ | 1280 |
| Crystal size | 0.21 × 0.21 × 0.21 mm |
| Theta range for data collection | 3.31 to 64.60° |
| Limiting indices | $-12 \leq h \leq 13$, $-11 \leq k \leq 8$, $-31 \leq l \leq 24$ |
| Reflections collected/unique | 12,552/4959 |
| Refinement method | Full-matrix least-squares on F$^2$ |
| Data/restraints/parameters | 4959/27/404 |
| Goodness-of-fit on F$^2$ | 1.716 |
| Final R indices [I > 2sigma(I)] | R1 = 0.1196, wR2 = 0.2847 |
| Largest diff. peak and hole | 0.720 and $-0.760$ e.Å$^{-3}$ |

In molecule A, the rings are planar. The RMSD of the ring C1A-C6A from the mean plane is 0.009(7) Å (atom C6A deviate by 0.003 Å from the mean plane defined for the ring). The RMSD of the ring C10A-C15A from the mean plane is 0.012(4) Å (atom C10A deviates by 0.009(4) Å from the mean plane defined for the ring). The phenyl rings are sp2 hybridized. The atoms C9A-O8A-C7A-C6A show torsion angles of $-6.6(7)°$, and suggests that they adopt +*anti*-clinal (+ac) conformation. Similarly, in molecule B, the rings are planar. The RMSD of the ring C1B-C6B from the mean plane is 0.006(5) Å (atom C6B deviates by 0.002(5) Å from the mean plane defined for the ring). The RMSDof the ring C10B-C15B from the mean plane is 0.013(4) Å (atoms C12B and C13B deviate by 0.012(4) Å from the mean plane defined for the ring). The phenyl rings are sp2 hybridized. The atoms C9B-O8B-C7B-C6B show torsion angle of $-3.9(7)°$, and suggests that they adopt +*anti*-clinal (+ac) conformation. Further, the structure is stabilized by C-H . . . O intermolecular hydrogen bond interactions. The details of the hydrogen bond geometry are given in Table 2. The molecule is reinforced by various π–π interactions. The π–π interactions exist between Cg2 and Cg4. Cg2 is the center of gravity of the phenyl ring (C10A-C15A) and Cg4 is the center of gravity of the ring C10B-C15B. The molecule exhibits medium to weak π–π interactions as the Cg2-Cg4 distance is 4.768(2) Å. The molecular packing showing C-H . . . O interactions is depicted in Figure 3.

**Table 2.** Hydrogen bond geometry(Å).

| D-H-A | D-H | H-A | D-A | D-H-A (°) | Symmetric Code |
|---|---|---|---|---|---|
| C22A-H22C-O20A | 0.96 | 2.33 | 2.736(5) | 104 | Intramolecular interaction |
| C22B-H22D-O20B | 0.96 | 2.33 | 2.721(6) | 103 | Intramolecular interaction |
| C9A-H23B-O20A | 0.96 | 2.17 | 2.911(6) | 133 | 1-x, −y, 1-z |
| C9B-H23D-O20B | 0.96 | 2.28 | 2.924(6) | 124 | −x, 2-y, 1-z |

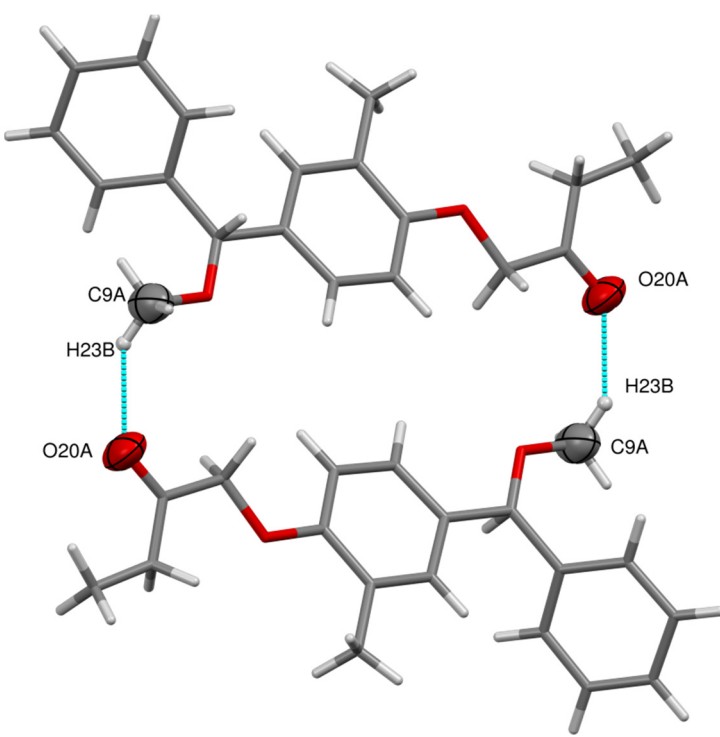

**Figure 3.** Molecular packing showing C-H . . . O interactions.

### 3.2. Effect of Crystal Compound on α-Glucosidase Diabetic Enzyme Inhibition and Kinetics

The crystal compound inhibited the α-glucosidase enzyme (IC$_{50}$: 10.30 ± 0.25 μg/mL). Acarbose (positive control) showed IC$_{50}$ values of 12.00 ± 0.10 μg/mL under the same conditions, indicating that the crystal compound inhibition occurred at considerably higher ($p \leq 0.05$) IC$_{50}$ values than with acarbose. To elucidate the manner of α-glucosidase inhibition, a kinetic analysis of the crystal compound was carried out by incubating it with diverse doses of $p$NPG (0.25–4 mmol L$^{-1}$ in the absence (control) or presence of the crystal compound at IC$_{20}$, IC$_{40}$, and IC$_{60}$ inhibitory concentrations (μgmL$^{-1}$)). Lineweaver Burk (LB) plots in the reaction were used to define the type of inhibition, as well as the Vmax and Km values. Figure 4 presents the LB plots of the crystal compound against the inhibition of α-glucosidase. Other than the various slopes and x-intercepts, the LB plots demonstrated that the intersecting point for diverse concentrations of the crystal compound came from the same y-intercept as the uninhibited enzyme. The slope and vertical axis intercept rose as crystal compound concentrations increased, with a corresponding increase in the horizontal axis intercept (−1/Km). The kinetic data indicated that with increasing concentrations of the crystal compound, the maximum velocity (Vmax) catalyzed by α-glucosidase remained constant. These findings suggest that the mechanism of α-glucosidase inhibition was reversible, and that it followed the conventional pattern of competitive inhibition. Dixon plots revealed that the inhibitory constant (Ki) for α-glucosidase was 0.41.

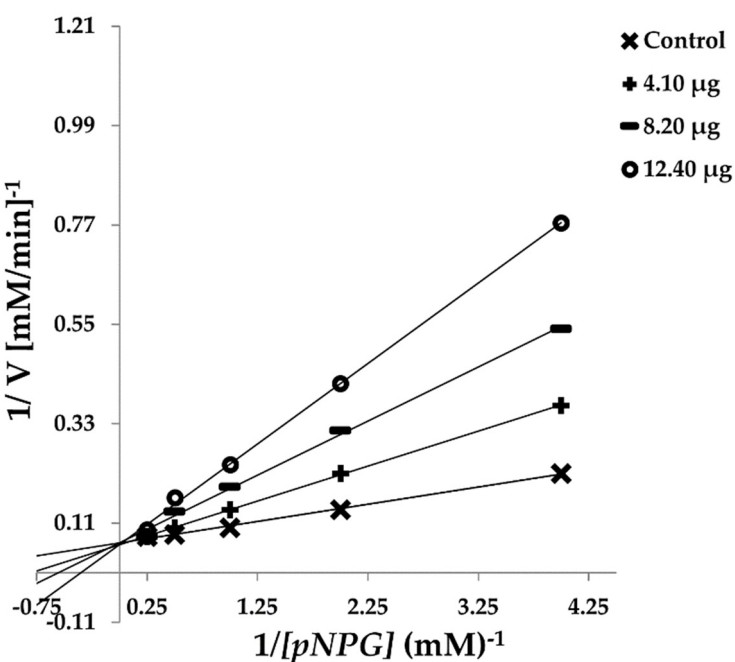

**Figure 4.** Lineweaver–Burk plot showing substrate-dependent enzyme kinetics when the crystal compound inhibits α-glucosidase enzyme.

### 3.3. Molecular Docking Simulation

Molecular docking simulation is used to determine how ligands interact with target proteins on a molecular level. It determines the degree of ligand binding, which indicates whether or not the protein is inhibited or activated. While binding with the inhibitor binding site of α-glucosidase, the molecule was found to be docked deep inside the binding pocket, occupying the cleft present in the active site. The compound was predicted to form a total of seven non-bonding interactions, including two hydrogen bonds with GLN 350 (2.82 Å) and ARG 312 (2.57 Å). The molecule also formed two hydrophobic π–π interactions with PHE 157 (3.98), HIS 239 (4.33) and a pi-alkyl bond with ARG 312 (4.93 Å). In addition, the molecule formed electrostatic π-anion and π-cation interactions with ASP 408 (4.98 Å) and HIS 239 (4.99 Å), respectively. With these interactions, the molecule had a binding affinity of −10.8 kcal/mol. However, acarbose was not able to bind with a higher binding affinity. It was also found that acarbose was not able to bind to the deep cleft of the active site, as the molecule did. Acarbose formed seven non-bonding interactions, all of them being hydrogen bonds. They included GLU 304 (2.36 Å), THR 307 (2.55 Å), SER 308 (1.92 Å), PRO 309 (1.77 Å), HIS 279 (2.49 Å), HIS 239 (3.02 Å), and PHE 157 (3.56 Å). With these interactions, acarbose had a binding affinity of −8.2 kcal/mol. Figure 5 depicts the binding interactions of the molecule and acarbose with the α-glucosidase. The outcomes from the docking simulation depict that the molecule can bind within the binding site of the enzyme and can induce biological activity, as was observed in the in vitro studies. According to Patil et al. [18] and Maradesha et al. [17], the docking was accurate, and the binding interactions were similar. The binding interactions validate the results obtained from XRD analysis. This shows that the C-H . . . O interaction is also observed by docking analysis. The intermolecular C-H . . . O hydrogen interaction which is observed in XRD analysis is also seen on the Hirshfeld surface map. The presence of C-H . . . O interaction in the molecule was confirmed the by the XRD analysis, Hirshfeld surface analysis and molecular electrostatic interactions.

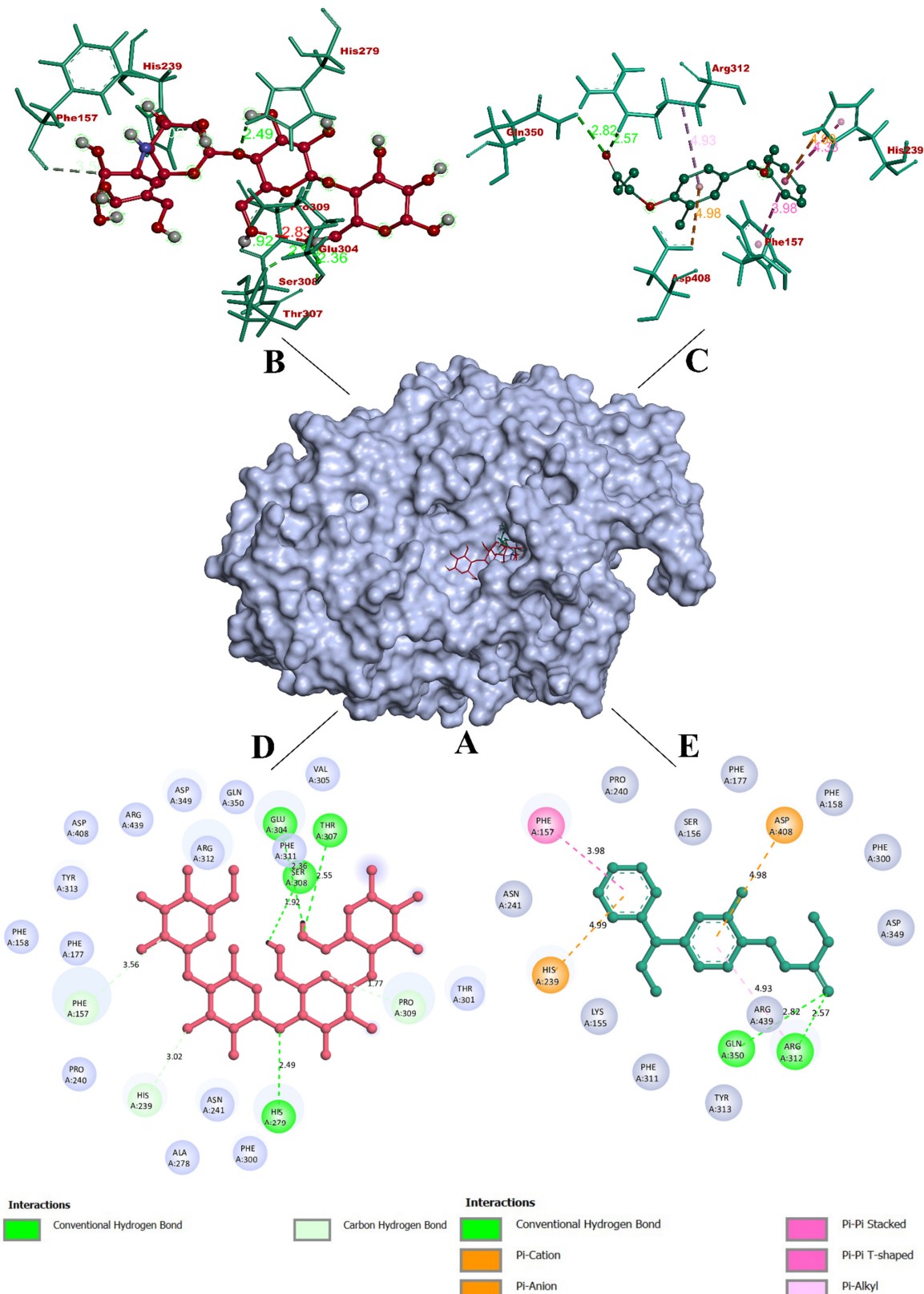

**Figure 5.** Visualization of binding interactions of the crystal compound and acarbose with the α-glucosidase; (**A**) surface diagram showing bound acarbose (red) and molecule (green); (**B**,**C**) 3D representation of acarbose (red) and molecule (green), respectively; (**D**,**E**) 2D representation of acarbose (red) and molecule (green), respectively.

### 3.4. Molecular Dynamics Simulation

The overall stability of the protein–ligand combination kept in a particular environment for a specific period of time was assessed using molecular dynamics simulation. Several metrics were evaluated during molecular dynamics simulations to determine the complex's overall stability, including the protein–ligand complex's RMSD, RMSF, Rg, SASA, ligand RMSD and ligand–hydrogen bonds. Over the course of a 100 ns simulation, the RMSD plot of the protein–ligand combination depicts the ligand's stability inside the binding pocket. On the other hand, the RMSF of a protein–ligand complex is used to calculate the average deviation of a particle (e.g., a protein residue) over time from a reference site. As a result, RMSF concentrates on the protein structural regions that differ the most from the mean. Furthermore, by calculating the root-mean-square distances with respect to the central axis of rotation, the radius of gyration (Rg) reflects the structural compactness of the molecules. For all protein–ligand complexes, SASA plots showed the area around the hydrophobic core generated between them. Only a few bonds were simultaneously broken and re-established during the simulation, with the majority of H-bonds remaining consistent with molecular docking. As a result, in dynamic trajectory analysis, ligand–hydrogen bonds are also important.

The RMSD plots depict that during molecular dynamics simulation, both the protein backbone atoms and the protein–crystal compound had a concurrent equilibration point, ranging from 0.30 to 0.35 nm, whereas the protein–acarbose plot was predicted with the RMSD value of 0.3 nm. In case of RMSF, the protein–acarbose plot was predicted with more fluctuations in comparison with the protein–molecule plot. All the plots were found with N-terminal, C-terminal, and loop fluctuations. In case of the Rg, the protein–acarbose and protein–molecule plots were both equilibrated within a range of 2.25–2.50 nm. In addition, the protein backbone atoms were found within a range of 3.0–3.25 nm. A similar pattern of results was obtained in the case of the SASA plots. The protein backbone atoms were predicted with a SASA value of 350 nm$^2$, whereas the protein–acarbose and the protein–molecule plots were both found within a SASA value range of 225–250 nm$^2$. During the ligand–hydrogen bond analysis, the molecule was found to have more hydrogen bonds (9), in comparison with the acarbose (3). It can therefore be asserted that the MD simulation results support the docking simulation outcomes. Visualizations of the MD trajectories are depicted in Figure 6. The outcomes of the MD simulation detail the overall stability of the molecule over the acarbose control during the 100 ns long simulation period. The concurrent plots of the protein–molecule complex with the protein backbone atoms indicate stronger binding affinity during the simulation study. The results obtained in this study are in accordance with previous studies in which MD simulation was performed for α-glucosidase [27,28].

### 3.5. Binding Free Energy Calculations

The binding free energy calculations revealed that the α-glucosidase–molecule complex has a better binding efficiency than the α-glucosidase–acarbose complex. Both the compounds used mainly Van der Waals energy to form the complex, followed by binding energy. The α-glucosidase–crystal compound complex had the highest amount of Van der Waals binding free energy (−108.593 kJ/mol). In all types of binding free energy, the α-glucosidase–crystal compound complex had more binding free energy than the α-glucosidase–acarbose. Results obtained from binding free energy calculations support the outcomes of both the docking and the MD simulation in terms of binding efficiency (Table 3). In addition, these outcomes were on par with those in previous studies that performed binding free energy calculations for α-glucosidase [17,18].

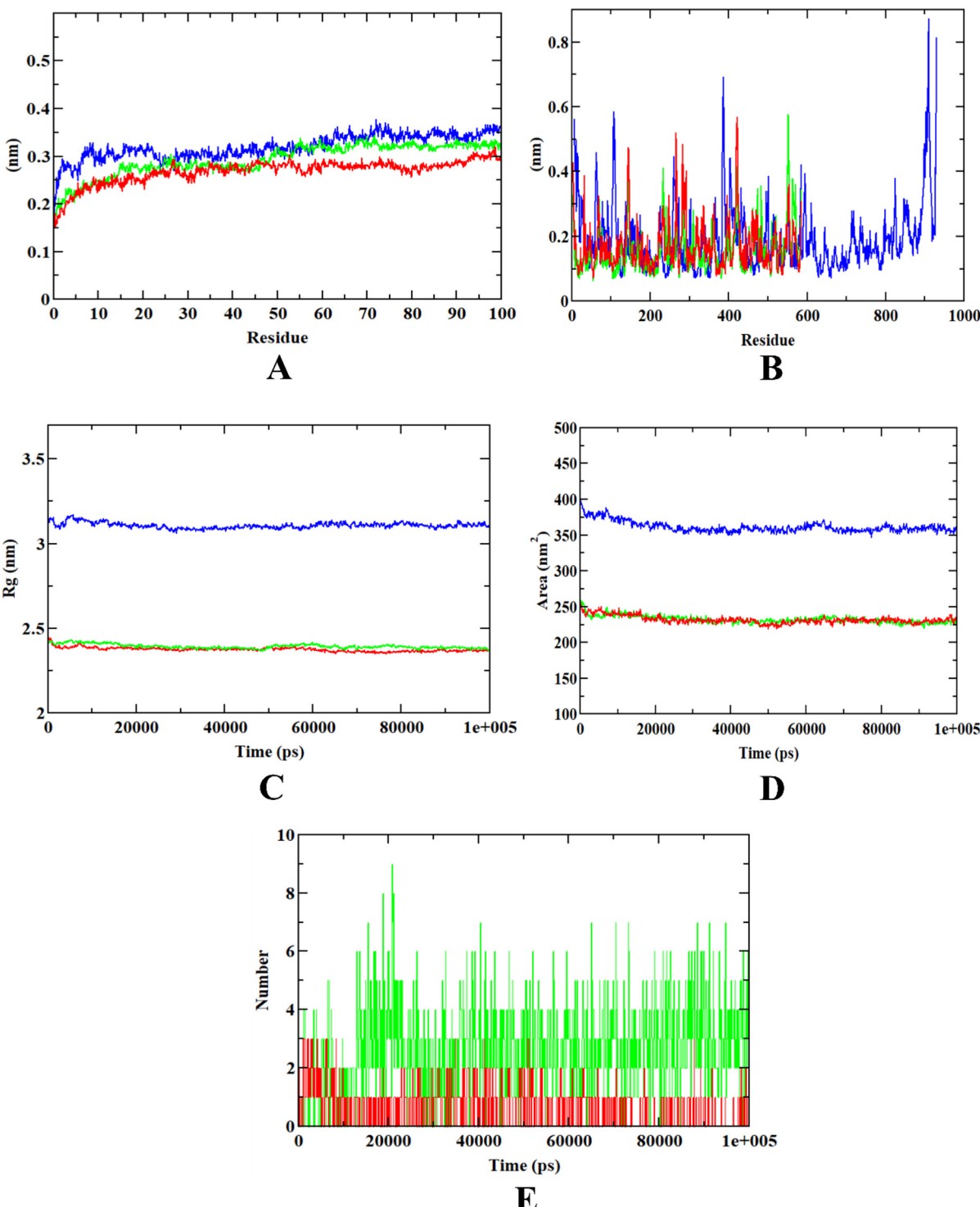

**Figure 6.** Visualization of the acarbose and crystal compound complexed with α-glucosidase MD simulation run for 100 ns; (**A**) RMSD; (**B**) RMSF; (**C**) Rg; (**D**) SASA; and (**E**) ligand hydrogen bonds. Blue: protein backbone atoms; red: protein–acarbose complex; green: protein–molecule complex.

**Table 3.** Binding free energy calculations for α-glucosidase–crystal compound complex and α-glucosidase–acarbose complexes.

| Types of Binding Free Energies | Values and Standard Deviations | Van Der Waal's Energy | Electrostatic Energy | Polar Solvation Energy | SASA Energy | Binding Energy |
|---|---|---|---|---|---|---|
| α-glucosidase-crystal compound complex | Values (kJ/mol) | −108.593 | −31.892 | 65.071 | −9.127 | −71.615 |
| | Standard deviation (kJ/mol) | ±12.178 | ±8.561 | ±9.298 | ±3.726 | ±10.028 |
| α-glucosidase-acarbose | Values (kJ/mol) | −99.716 | −27.716 | 49.918 | −7.561 | −61.239 |
| | Standard deviation (kJ/mol) | ±9.257 | ±7.145 | ±8.769 | ±2.539 | ±8.751 |

### 3.6. Hirshfeld Surface Analysis

Hirshfeld surface analysis is used to visualize the intermolecular interactions in a crystal. It is a powerful tool for analyzing intermolecular interactions, such as hydrogen bonds and C-H contacts [29]. These intermolecular interactions can be summarized in two-dimensional fingerprint plots. The distance from the nearest atoms inside and outside the Hirshfeld surface are characterized by the quantities $d_i$ and $d_e$, respectively. The normalized contact distance ($d_{norm}$) is

$$d_{norm} = (d_i - r_i^{vdW})/r_i^{vdW} + (d_e - r_e^{vdW})/r_e^{vdw} \quad (3)$$

where $r_i^{vdw}$ and $r_e^{vdw}$ are the van der Waals radii internal and external to the surface. The close intermolecular contacts are represented by red-colored regions on the Hirshfeld surface. The Hirshfeld surface was generated using CrystalExplorer17 software [30]. Figure 7 shows the Hirshfeld surface mapped with $d_{norm}$. The $d_{norm}$ surface is drawn in the range −0.40 to 1.70. The bright red regions on the Hirshfeld surface represent C-H . . . O intermolecular interaction. The fingerprint plot gives the atomic pair-wise interactions. The precise two-dimensional fingerprint plots are shown in Figure 8. The major contribution is from H-H (63.4%) contacts to the total Hirshfeld surface area and the least contribution is from O-H/H-O (14.5%) contacts. The remaining interactions observed are C-C and O-O and C-O/O-C, which contribute less to the Hirshfeld surface area. They do not show dominant contributions to the total surface.

### 3.7. Frontier Molecular Orbitals

The frontier molecular orbitals (FMOs) are the highest occupied molecular orbital (HOMO) and lowest unoccupied molecular orbital (LUMO). GAMESS software [31] was used to perform density functional theory (DFT) calculations. The energy levels of the FMOs were computed using the B3LYP/6-31 G (d, p) basis set and are displayed in Figure 9. The calculations show that the HOMO and LUMO are mainly localized on the benzene ring. The energy difference between the HOMO and the LUMO indicates the energy gap. The energy gap between the HOMO and the LUMO in this case was 4.952 eV. This energy difference predicts that the molecule is kinetically stable. As the value of the HOMO–LUMO gap increases, the molecule becomes less stable. A larger energy gap between the HOMO and the LUMO is associated with high kinetic stability [32]. The molecular descriptors calculated from the energy values of the frontier molecular orbitals are listed in Table 4.

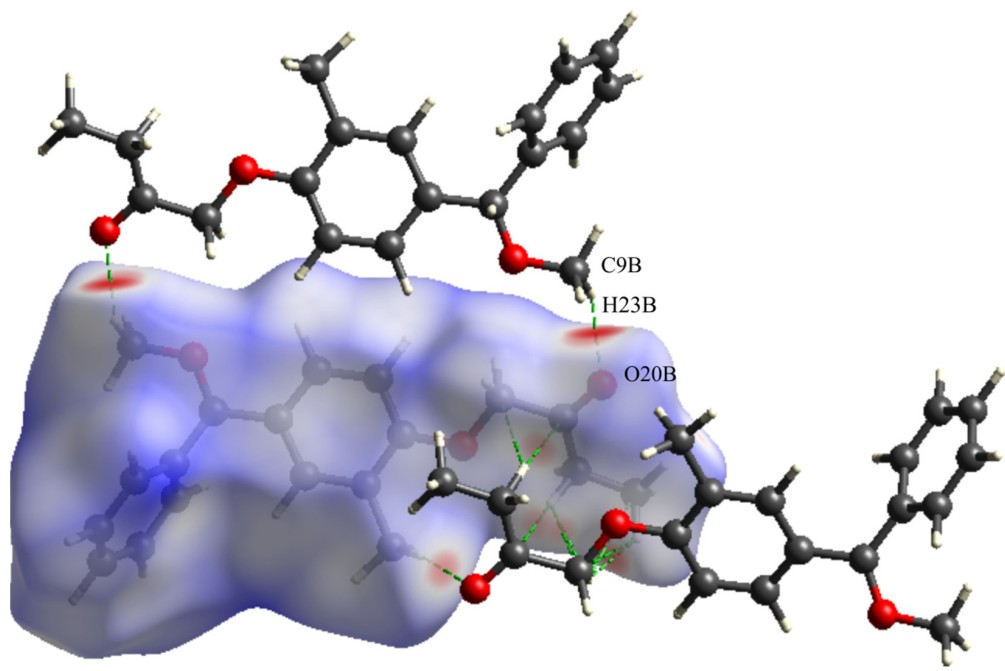

**Figure 7.** Hirshfeld surface mapped with $d_{\mathrm{norm}}$.

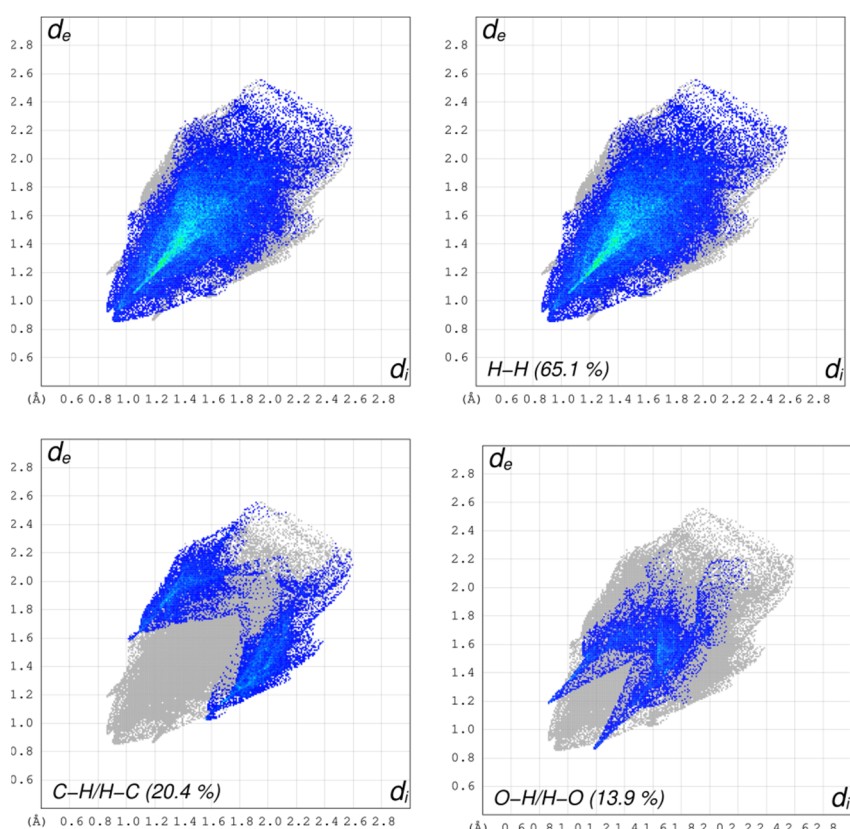

**Figure 8.** Fingerprint plots for the title molecule.

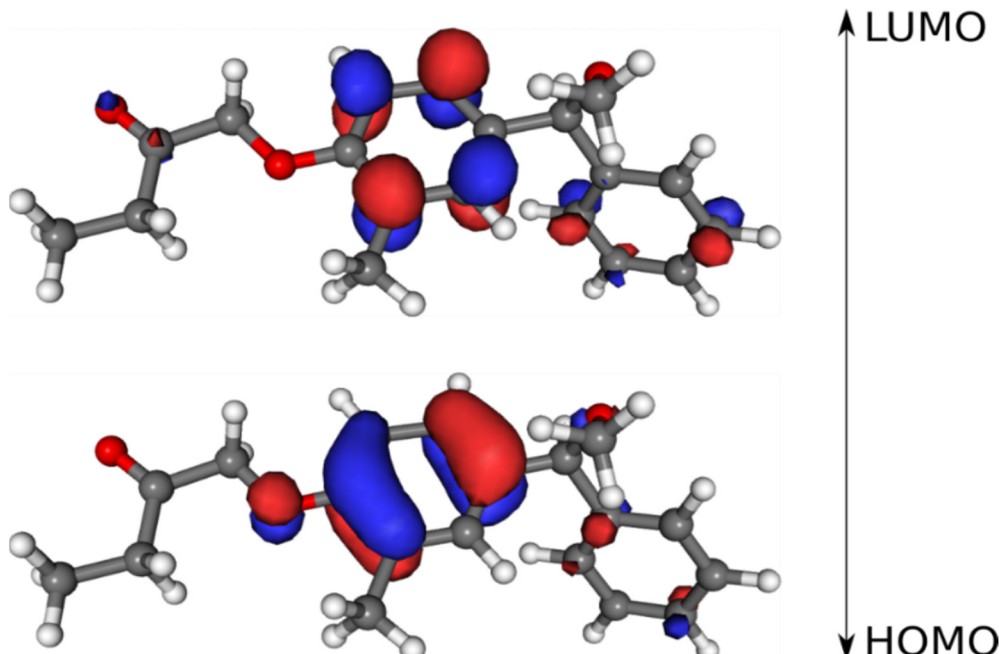

**Figure 9.** Energy levels of the frontier molecular orbital of the title molecule.

**Table 4.** Calculated energy values of molecular descriptors of the title compound.

| Molecular Descriptor | Value |
|---|---|
| HOMO | −8.489 eV |
| LUMO | 3.537 eV |
| Energy gap (ΔE) | 4.952 eV |
| Ionization potential (I) | 8.489 eV |
| Electron affinity (E) | −3.537 eV |
| Chemical potential (μ) | −6.013 eV |
| Electronegativity (χ) | 6.013 eV |
| Global hardness (σ) | 2.476 eV |
| Global softness (η) | $0.2019 \text{ eV}^{-1}$ |
| Electrophilicity (ω) | 7.301 eV |

### 3.8. Molecular Electrostatic Potential (MEP)

The molecular electrostatic potential (MEP) map is a tool to analyze the charge distribution in the molecule. The MEP surface was generated using the Gaussian09 program [33] and is shown in Figure 10. The molecular electrostatic potential is represented by different colors. The value of potential increases, with red < orange < yellow < blue. The deep red areas indicate regions with a negative electrostatic potential and the blue sites indicate regions with a positive electrostatic potential. Figure 10 shows that an electropositive region (blue) is observed around the hydrogen atom and the negative region (red) is concentrated over the oxygen atom, indicating an electrophilic area.

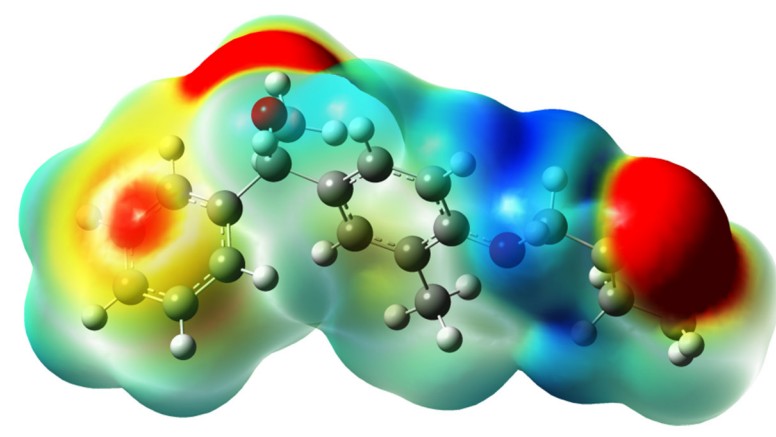

-2.6e-2 ▬▬▬▬▬ 2.6e-2

**Figure 10.** Molecular electrostatic map of the compound.

## 4. Conclusions

The crystal compound was synthesized and the single crystal developed by means of the slow solvent evaporation technique using ethanol as a solvent. The compound was characterized using NMR and mass spectroscopy, and, finally, the molecular structure of the compound was confirmed by single crystal X-ray diffraction. In vitro α-glucosidase inhibition for the crystal compound was better than with the acarbose. A molecular docking study revealed that the probable binding interaction of molecule (3) with the protein target showed a low binding free energy, which prompted us to look for further experimental studies. The outcomes from in silico studies depicted that the molecule was able to inhibit the enzyme by interacting with the binding pocket residues. The molecule was stable throughout the simulation run of 100 ns, indicating that it can efficiently carry out the biological activity of enzyme inhibition The energy difference between the frontier molecular was 4.95 eV, which predicts the title molecule is kinetically stable. The molecular electrostatic potential surface revealed the electronegative and electropositive sites present around the oxygen and hydrogen atoms, respectively.

**Author Contributions:** Planning and conceptualization of the manuscript: L.R. and R.R.; data analysis and method development: S.A.K., E.S. and C.M.; writing: preparation of original draft and writing—V.L.R., R.R., L.R.N., P.A., V.S., R.R.A. and M.A.S.; supervision and editing—C.S. and S.M.P. All authors have read and agreed to the published version of the manuscript.

**Funding:** This research received no external funding.

**Institutional Review Board Statement:** Not applicable.

**Informed Consent Statement:** Not applicable.

**Data Availability Statement:** Not applicable.

**Acknowledgments:** Chandra and Lakshmi Ranganatha V gratefully acknowledge the principal and NIE-Management for the support and encouragement to carry out this research. Additionally, Akhileshwari P. thanks DST-KSTePS, Government of Karnataka, Bengaluru for providing the fellowship. All the authors thank JSS Academy of Higher Education and Research (Mysore, Karnataka, India) for their kind support, encouragement, and provision of the necessary facilities.

**Conflicts of Interest:** All authors declare no conflict of interest.

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
