# Peer review of "Synthesis, Characterization, Hirshfeld Surface Analysis, Crystal Structure and Molecular Modeling Studies of 1-(4-(Methoxy(phenyl)methyl)-2-methylphenoxy)butan-2-one Derivative as a Novel α-Glucosidase Inhibitor"

_crystals, doi:10.3390/cryst12070960_

Round 1

Reviewer 1 Report

Chandra , Mallikarjunaswamy C et al reported Synthesis, Characterization, Hirshfeld Surface Analysis, Crys- 2 tal Structure and Molecular Modeling Studies of 1-(4-(meth- 3 oxy(phenyl)methyl)-2-methylphenoxy)butan-2-one Derivatives 4 as Novel α-Glucosidase Inhibitors. In my point of view manuscript needs major revision.

1- Introduction must rewritten againt eith significant literature survey.

2- In theoriritical part  what programme used? what basis set? what method?

3- what is importance of theoritical part? if there is any cobination with experimental part?

4- The figure of MEP is not clear!!! where is green and blue regions??

5- In Molecular Docking Simulation the author must demostrate why selection of this protein?

6- In Hirshfield Surface Analysis, more discussion is needed.

Reviewer 2 Report

Reviewer comments for crystals-1749285

This study reports the synthesis, crystal structure, and molecular docking of a phenoxy ketone compound. Some results reported in this manuscript are interesting. However, some improvements and clarifications are required prior to publication in Crystals.

Major issues:

1.      Title, abstract, and keywords

1.1.      The title must be changed from “Derivatives as Novel α-Glucosidase Inhibitors” to “Derivative as Novel α-Glucosidase Inhibitor”  since only one compound is studied and reported.

1.2.      What is the significance of stating lattice parameters in the abstract section?

1.3.      A brief detail regarding the essential structural features is needed to be found in the abstract section.

1.4.      The current abstract does not pinpoint the importance of the work.

1.5.      Methoxy acetate must not be listed as a keyword.

2.      Introduction section

2.1.      The current introduction is too vague. The authors describe the modification of phenoxy acetates to get some derivatives with biological activities. However, they are not significantly related to the compound synthesized and investigated in this study. Therefore, the introduction section must be rewritten to show the state-of-the-art for the potential compounds with more structurally related.

3.      Experimental section

3.1.      Please state the purity of each chemical used.

3.2.      According to the synthetic scheme (Figure 1) of the title compound (3), why do the authors use the molar ratio of compound (1) to compound (2) = 0.01 to 0.02?

3.3.      There are some updated versions of the SHELX package. Why do the authors use SHELXS/L-97 in their structural solution and refinement process?

3.4.      More infrared peaks are needed to be listed.

3.5.      What is the meaning of “bm” in the 1H-NMR data?

4.      Results and discussion section

4.1.      According to the crystal structure reported in Figure 1, why do the authors skip to label C1 in their structure refinement? Please refine again and begin with C1.

4.2.      Figure 2(b) is meaningless. Please redraw the crystal packing to highlight the significant interactions within the crystal of the compound (3) and describe the interactions observed after the description of the structural feature. Please also add a table and/or a figure caption that lists or indicates the essential intermolecular interactions.

4.3.      CheckCIF report contains several Alert B. Please describe in detail the leftover of each Alert B.

4.4.      PLAT883_ALERT_1_G No Info/Value for _atom_sites_solution_primary. Please Do.

4.5.      PLAT965_ALERT_2_G The SHELXL WEIGHT Optimisation has not Converged. Please Check.

4.6.      In Figure 6, the atomic label for atoms that are close to the faint red spots must be shown.

4.7.      The dnorm range of the Hirshfeld surface must be given in the manuscript.

4.8.      Some variables in the dnorm equation are not yet defined.

4.9.      In Figure 7, the picture on the top left and right should be the other way round. Please also add the description to the fingerprint plot that shows all interactions.

4.10.   The authors should describe the relations of the results obtained from the analyses of Hirshfeld surface, frontier molecular orbitals, and molecular electrostatic potential to the binding characteristics toward the protein.

Minor comments:

1.     Numerous grammatical errors are found throughout the manuscript. For instance:

-       (Page 1, Line 33) “frontier molecular orbital” must be changed to “frontier molecular orbitals”

-       (Page 2, Line 53) “novel compounds” must be changed to “novel compound”.

-       (Page 4, Line 134) “[with” must be changed to “with”.

-       (Page 14, Line 313) “the  packing” has extra space.

-       (Page 14, Line 320) “X-ray diffraction studies” must be changed to “X-ray diffraction study”.

2.     Formatting errors are also found. For example:

-       “CH3” must be changed to “CH3

-       “dnorm” must be changed to “dnorm

-       Citing researchers from literature does not need to use bold letters.

-       The format of references is not consistent. Some use commas to separate authors, but others use semicolons.

3.     References are too few and most of them are not up to date.

Round 2

Reviewer 1 Report

In theoretical part what programme used? what basis set? what method?

Authors’ response: As per the reviewer’s suggestion, details of the used softwares have been added in the “Materials and Methods” section. The GAMESS software was used for calculations with B3LYP/6-31 G (d, p) basis set.

I can nof find it

 The figure of MEP is not clear!!! where is green and blue regions?

Authors’ response: As per the reviewer’s suggestion, Figure of MEP is changed. To avoid further confusions, green and blue regions were not highlighted in the given image.

It still unclear

Reviewer 2 Report

Reviewer comments for crystals-1749285R1

Critical defects remain in the revised manuscript, so I recommend further revision before publication in Crystals.

1.      (Previous comment) What is the significance of stating lattice parameters in the abstract section?
Authors’ response: As per the reviewer’s suggestion, significance of stating lattice parameters have been added in the abstract section.
- It is not found in the revised manuscript.

2.      (Page 6, Lines 208-209) “The details of hydrogen bond geometry is given in table xx.
Please assign the table number

3.   (Page 6, Line 213) “α=30.1(2)°, β = 52.2°, and γ = 30.3°” How these lattice parameters are not related to the intermolecular interactions. Why do the authors mention them here? Moreover, they are not associated with the information in the Abstract section and Table 2. 

4.   (Page 6, Lines 213-214) “Molecular packing showing C-H…O interactions is depicted in figure xx.” Please assign the figure number.

5.    Figures 2(B) and 3 must be improved. There are many irrelevant interactions found in both the figures. Moreover, the Figures 2(B) and 3 must be related to Table 2 (No illustration regarding the intramolecular interactions).

6.      The dnorm range must be given in two decimal places.

7.    The authors should carefully read the previous comment that I did not recommend changing “dnorm” to “d”. Therefore, please change “d” to “dnorm” in the Hirshfeld section.

8.      The numbering labels for carbon, hydrogen, and oxygen are recommended to be added to Figure 6 for better understanding.

9.     In this revised manuscript, numerous typoes and grammatical mistakes (too much to be able to list) have been found. Please deliberately recheck your manuscript before the next submission.   
